# A Systematic Review of the Basis for WHO’s New Recommendation for Limiting Aircraft Noise Annoyance

**DOI:** 10.3390/ijerph15122717

**Published:** 2018-12-02

**Authors:** Truls Gjestland

**Affiliations:** SINTEF DIGITAL, N-7465 Trondheim, Norway; truls.gjestland@sintef.no; Tel.: +47-932-05-516

**Keywords:** aircraft noise, annoyance, dose–response, environment, WHO guidelines

## Abstract

The new WHO Environmental Noise Guidelines for the European Region have recommendations for limiting noise exposure associated with adverse health effects. The limits are said to be based on a systematic review of existing evidence. This paper gives a systematic assessment of the presented evidence with respect to annoyance from aircraft noise. The new guidelines have been based on the results from a selection of existing aircraft noise studies. This paper demonstrates that a similar selection of other existing post-2000 studies will yield very different results. In addition, the validity of the presented evidence has been questioned as some of the referenced studies have not been conducted according to standardized methods, and the selection of respondents is not representative of the general airport population.

## 1. Introduction

The World Health Organization just recently published their report Environmental Noise Guidelines for the European Region [1]. The report is the result of the joint effort by a team of researchers covering all aspects of negative health effects by exposure to environmental noise. According to WHO, the main purpose of these guidelines is to provide recommendations for protecting human health from exposure to environmental noise originating from various sources.

WHO stresses that all reasonable precautions have been taken to verify the information contained in the publication. However, the published material is being distributed without warranty of any kind from WHO, and to be on the safe side, they state that the responsibility for the interpretation and use of the material lies with the reader.

This reservation is very appropriate regarding the guideline’s chapter on aircraft noise annoyance. In the new guidelines WHO strongly recommends “reducing noise levels produced by aircraft below **45 dB** L_den_, as aircraft noise above this level is associated with adverse health effects”.

This recommendation is based on the idealistic assumption that nobody should ever be exposed to noise levels which endanger complete individual well-being or quality of life, and, as such, it is useless for general regulatory purposes. Nevertheless, the recommendation will be observed with great interest by individuals and groups advocating reduced noise exposure from aviation. It is therefore unfortunate that the recommendation is based on a specific set of data whose choice has a great impact on the proposed recommendations.

## 2. Systematic Review of Evidence

### 2.1. WHO Dataset

Several groups of researchers were commissioned by WHO to compile results from recent surveys on health effects of noise. The group of researchers who worked on the impact of environmental noise on annoyance presented a systematic review of studies that had been published during the time period 2000–2014 [2]. Guski and his coauthors had developed a strict protocol for selection of studies. The inclusion criteria comprised inter alia:Participants should be members of the general populationAnnoyance question and response format should follow (as close as possible) recommendation given by ICBEN and/or ISO TS 15666

The authors went through an extensive search in existing databases and came up with a list of 15 aircraft noise annoyance studies that complied with their inclusion criteria. After an additional elimination process 12 studies were selected for the final meta-analysis. For the three excluded studies the authors could not find a regression function that they could use to estimate % HA. 

One of these studies comprises surveys at two different airports, but Guski et al. [2] only considered one of them for reasons unknown. It is interesting to note that results from previous similar surveys at both of these airports which were excluded, were included in the analysis by Miedema & Vos [3] for their well-known EU reference curve. The excluded survey was analyzed according to the CTL (Community Tolerance Level) method which yields a dose–response curve for the full exposure range. However, Guski et al. seem to be unfamiliar with the standard ISO 1996, Annex E.

The final list of candidate studies on aircraft noise annoyance for their meta-analyses is shown in Table 1. They called the results from these 12 studies WHO full dataset. These surveys were conducted during the period 2001–2011. The list comprises data from a total of 17,094 respondents. 

The table contains information on the airports and their respective IATA codes for identification, reference to the publication of the survey results, total number of respondents per survey, calculated Community Tolerance Level (CTL). The Community Tolerance Level is defined in the standard ISO 1996-1. The CTL value is a single-number quantity that defines a unique relationship between noise exposure and the percentage of the exposed population that is highly annoyed [10], and a classification “rate of change”, H/L, (that will be explained later). 

Guski et al. offered the following scatterplot and quadratic regression of the relationship between aircraft noise, L_den_, and the prevalence of highly annoyed residents, % HA (highly annoyed), Figure 1.

The data points in Figure 1 do not represent aggregated empirical observations as is usual for such plots. They represent predicted values estimated from the regression equations for each of the studies. Different regression models have been used in the respective studies, and the regressions have been based on different exposure ranges. Finally, the results for the WHO full dataset have been found using a quadratic regression model and weighting according to study sample size.

The procedure of applying a regression model to data points derived from other (and different) regression models makes it almost impossible to assess the confidence interval for the final curve.

A procedure based on combining all responses from different surveys in this manner represents simple way of analyzing data from aircraft noise annoyance surveys. It ignores the fact that only about one third of the variance in the response data is explained by the cumulative noise exposure [11] and it effectively prohibits any possibility of studying the influence of non-acoustic factors; an issue that has received an emerging and growing interest.

A visual inspection of the data in Figure 1 shows that for the noise exposure range of most practical interest for regulatory purposes, L_den_ 50 dB to L_den_ 60 dB, the prevalence of highly annoyed residents varies between approximately 5% and 70%. It is difficult to attribute this enormous spread to personal or situational attitudes towards the cumulative noise exposure only. A more plausible explanation would be that there must be other factors that also play an important role. This fact is not commented on and completely overlooked by the researchers responsible for the presentation of evidence for the WHO guidelines. 

The WHO Guidelines Development Group also seems to be ignorant about the importance of non-acoustic factors. In the Guidelines publication it is stated that “in noise annoyance studies non-acoustic factors may explain up to 33% of the variance” [1] (p. 14). This must be a misunderstanding. The correct statement should be that acoustic factors (or rather L_den_ -based factors) may explain up to 33% of the variance, while the other two-thirds are explained by non-acoustic factors.

### 2.2. The HYENA Study

The results from surveys at six airports from the HYENA study have been included in the WHO full dataset. These survey results have been reported by Babisch et al. [4], see Table 1. The HYENA study was primarily designed to study hypertension among residents near airports and included respondents aged 45–70 years only. Most surveys have respondents aged 18 years and up. This is for instance the case for the 20 studies that are included in the Miedema & Vos curve, and which has become a de facto EU standard reference curve for aircraft noise annoyance [3]. The annoyance response is age-dependent with a maximum sensitivity around 45 years as reported by Van Gerven et al. [12]. These authors found that for aircraft noise at L_den_ 55 dB the prevalence of highly annoyed persons was about 25% among people aged 20 years and significantly higher, 43%, among people 45 years old. According to their analysis the difference in the annoyance response in a group of respondents evenly distributed across an age span 20 to 80 years compared to a similar group 40 to 70 years is about five percentage points. 

Guski et al. are aware that this fact has most certainly contributed to an increase in annoyance in the HYENA study, but still they choose to include the data in violation of their own selection criteria (“member of the general population”). If the HYENA results follow the general trend, a certain bias towards higher annoyance must be expected. The HYENA results comprise 28 percent of the WHO full dataset.

Another selection criterion was that the annoyance question and the response format should follow the recommendations given by ICBEN (International Commission on Biological Effects of Noise) [13] and/or ISO TS 15666 [14] or at least be very similar. These recommendations specify an annoyance question without mentioning any particular time-of-day. The HYENA study, however, had two separate questions on “annoyance due to aircraft noise during the day” and “…during the night”. This fact has been commented upon by the authors, but they conclude that the response to the daytime period can be used in their analysis, again a violation of their own inclusion criteria. This decision may be disputed. Upon request, Guski et al. justify their decision by citing the HYENA report [4] (p. 1174). The authors of this report “assume that the overall annoyance (day + night) is mostly determined by the annoyance during the daytime”.

However, the question used in the HYENA study can be interpreted in different ways, for instance: How annoyed are you in general by aircraft noise during the day?How annoyed are you during the day by aircraft noise during the day?How annoyed are you during the day by aircraft noise in general?

One cannot assume that all three interpretations yield the same response, and identical to the response to the question: How annoyed are you in general by aircraft noise in general?

In the HYENA study the annoyance response during the night was lower than during the day, and one could assume that the annoyance in general, i.e., without referring to a certain period, would be somewhere between the two responses. However, in a study of noise around Cologne/Bonn airport Bartels [15] observed a higher prevalence of highly annoyed residents than what could be predicted from the Miedema & Vos curve. She attributed this to the relative high portion of night time flights. It is highly questionable to combine the responses from the HYENA study (annoyance during the day) with those that strictly follow the ICBEN recommendations (general annoyance with no specified time period). 

A visual inspection of the annoyance data from the HYENA study reveals that two airports, Athens (ATH) and Milan (MXP) have an exceptionally high prevalence of highly annoyed neighbors, see Figure 1. The field work for the Athens study was conducted in 2003, but this airport was not opened until March 2001. First, this fact is in violation with one of the selection criteria in the HYENA study, «persons living for at least 5 years near the airport», and secondly, someone that has endured a noisy construction period of perhaps 3 to 4 years and then suddenly has been exposed to unfamiliar aircraft noise for two years, cannot be considered a typical airport neighbor. 

There are two major airports serving the city of Milan, Malpensa (MXP), and Linate (LIN). Two years prior to the survey at Malpensa, Linate airport experienced a tragic crash with 114 casualties. This triggered a lot of public discussion about air traffic safety. This may very well have influenced the response at Malpensa. High fear of accidents has been found to shift the annoyance response equivalent to as much as 20 dB in the exposure [16,17]. Milan Malpensa can therefore hardly be considered representative for a typical European airport.

In a report on the results of the HYENA study Babisch et al. comment on the very high annoyance scoring of the Athens and Milan airports. They discuss several reasons for this and conclude that the data from these two airports is not representative for airports in general. They therefore exclude the data from their subsequent pooled analyses [4] (p. 1175). Nevertheless, Guski et al. include both airports in the WHO full dataset. 

### 2.3. Response Weighting

Analysts often rely on statistical software to develop regression-based dose–response relationships without detailed concerns for the assumptions made by various regression techniques, and for their implications. A typical pitfall is weighting according to the number of respondents. This may make perfect sense when studying a one-dimensional problem. However, this is not the case when analyzing aircraft noise annoyance. 

The annoyance response to aircraft noise is governed not only by the cumulative noise level but also by several other factors, both acoustical and non-acoustical [18]. According to Basner et al. [11] cumulative measures of noise exposure per se, expressed in units similar to Day-Night Average Sound Level (DNL), account for approximately one third of the variance in community-level data. 

A good estimate for the annoyance response in a certain noise situation can be found using a relatively small survey sample. An increase in the number of respondents will shrink the confidence interval around this estimate but will normally not change the value of the estimate significantly. Consider two airports that are very different with respect to non-acoustical factors. The results of an annoyance survey yield two different dose–response curves. If the task is to find an average dose–response curve for the two airports, one would intuitively assume that this curve would be located midway between the two initial ones. However, if a weighting corresponding to the number of respondents per survey is applied, the average response curve could be located anywhere between the two, and closest to the curve corresponding to the survey with the highest number of respondents. In other words, the result will be highly dependent on the study design (number of respondents per airport). A weighting of multidimensional responses according to the number of respondents is therefore not recommended.

Response weighting has been applied by Guski et al. for the WHO full dataset. Studies at Amsterdam airport comprise 6771 respondents equal to about 40 percent of this dataset. Any specific non-dose factor that may be present at this airport will therefore have a prominent and disproportionate influence on the final exposure–response function.

### 2.4. High-Rate and Low-Rate Airport Change Situation

Most airports experience an increase in traffic. This increase usually occurs gradually over many years. Other airports are characterized by large abrupt changes such as the opening of a new runway, introduction of new flight paths, an abrupt increase in number of aircraft movements, etc. 

Janssen and Guski [19] call airports low-rate change airports if there is no indication of a sustained abrupt change of aircraft movements, or the published intention of the airport to change the number of movements within three years before and after the annoyance study. They offer the following definition: An abrupt change is defined here as a significant deviation in the trend of aircraft movements from the trend typical for the airport. If the typical trend is disrupted significantly and permanent, we call this a ‘high-rate change airport’. We also classify this airport in the latter category if there has been public discussion about operational plans within (three) years before and after the study. Low-rate change is the default characterization.

Gelderblom et al. [20] have applied this “high-rate/low-rate” classification to 62 aircraft noise annoyance studies conducted over the past half century. They show that there is a difference in the annoyance response between the two types amounting to about 9 dB. To express a certain degree of annoyance people at a high-rate change (HRC) airport on average “tolerate” 9 dB less noise than people at a low-rate change (LRC) airport. Guski et al. [2] report a similar but somewhat smaller, 6 dB, difference. Any attempt to develop an average dose–response curve from at set of studies will therefore be highly dependent on the types of airports that are included. A high percentage of HRC airports will increase the average prevalence of highly annoyed people.

Guski et al. [2] have done a characterization of the 12 studies included in the WHO full dataset. This is shown in Table 1. They have not done any assessment of Zurich and Milan airports. They state there is “a tendency in the direction HRC” but find that these two airports do not fit exactly to the definition. In our opinion they are clearly HRC airports. There have been long-lasting public discussions about flight routes in Zurich. At Milan Malpensa the traffic volume almost tripled in late 1998 when Alitalia moved their major hub to this airport. This was a little more than four years prior to the survey. The above-mentioned tragic accident at Milan Linate only two years before the survey may also have contributed to a high annoyance response. We are inclined to categorize both Zurich and Malpensa as HRC.

In 2009 the decision to expand the Hanoi Noi Bai Airport had already been made, and the public knew there would be an increase in traffic. The new terminal was opened in 2014 causing a 30% increase in the traffic volume. In our opinion Hanoi Noi Bai is a “borderline HRC” airport. 

If these three airports—ZHR, MXP, and HAN—are also included in the HRC category, the WHO full dataset comprises eight out of 12 HRC airports or about to 83% of the respondents. In contrast, in the dataset presented by Gelderblom et al. 17 out of 62 airports or about 35% of the respondents have been categorized as HRC, and in the original dataset used by Miedema and Vos for their dose–response curve [3] only two out of 20 airports or about 10% of the respondents were categorized as HRC.

### 2.5. CTL Analysis

The CTL method [10] provides an accurate and convenient way of comparing the results from different annoyance surveys. The CTL value is a single number quantity that characterizes the results of a single survey or a set of surveys. Each CTL value is associated with a complete dose–response curve.

The average CTL value for the 12 studies included in the WHO dataset is L_CT_ 66.1 dB with a standard deviation of ±6 dB. It should be noted that this calculation is based on some results from surveys not conducted according to standardized methods (The HYENA study). Figure 2 shows the average CTL curve and the dose–response function presented by Guski et al. [2]. In addition, the EU reference curve by Miedema and Vos [3] has been plotted for comparison. The CTL value corresponding to the EU reference curve is L_CT_ 73.7 dB [20]. The figure indicates an increased prevalence of highly annoyed residents in the WHO full dataset, equal to a shift of 7.6 dB towards lower noise levels. In other words: people included in the WHO full dataset seem to “tolerate” 7.6 dB less noise than what was observed by Miedema and Vos in order to express a certain degree of annoyance.

### 2.6. Alternative Selection of Surveys

A literature search for post-2000 aircraft noise annoyance surveys yields 18 surveys that adhere to the inclusion protocol defined by Guski et al. [2] and for which we have sufficient data to do a comparative analysis. Six of these were included in the WHO full dataset. There are also reports from other surveys, but their design deviate too much to be readily included. The list of surveys comprises 12 studies in Europe, five studies in Asia, and one in the US. These surveys are listed in Table 2.

The selection of surveys comprises 16,047 individual participants. Half of the airports are categorized HRC airports and these comprise ~60% of the respondents. The average unweighted CTL value for these surveys is L_CT_ 70.7 ± 7 dB. The corresponding dose-response curve can be calculated as described in the standard ISO 1996-1, Annex E [10].

This curve is shown in Figure 3 together with the EU reference curve [3]. The average response lies above the reference curve, indicating a higher prevalence of annoyance. However, the difference between the two curves is less than 1 σ (one standard deviation). Their CTL values differ by only 3 dB; therefore, one cannot conclude that they are significantly different.

The WHO recommends that the noise is kept below a level corresponding to 10% highly annoyed. For this alternative set of survey data 10% HA corresponds to exposure to aircraft noise at L_den_ 53.4 dB, in other words substantially higher than the guideline value L_den_ 45 dB. 

## 3. Conclusions

The recommendations regarding aircraft noise annoyance in the new WHO Guidelines for Environmental Noise [1] are based on noise surveys conducted after 2000. A set of surveys was selected and analyzed by a team of researchers commissioned by WHO. 

This paper demonstrates that the selection of surveys and the method for analyzing the results have a huge impact on the final recommendations.

The respondents in half of the selected surveys were recruited from a specially noise sensitive age group not representative for the general airport population. In addition, the non-standardized questionnaire that was used may not give comparable annoyance results. Two surveys had exceptionally high annoyance scores and were discarded as outliers by the researchers that conducted them. Nevertheless, the results were included in the WHO full dataset. One particular airport contributed 40 % of the data, thus giving this airport a disproportionate influence on the result. The team that collected the evidence assigned the grade “*moderate quality*” to their proposed dose-response function.

The *moderate quality* evidence report was used by the WHO Guidelines Development Group to *strongly recommend* a limit of L_den_ 45 dB to avoid adverse health effects from aircraft noise.

A separate dataset has been compiled from 18 post-2000 aircraft noise surveys. All of these surveys were conducted strictly in compliance with recommended standardized methods. The survey results were analysed according to the CTL method described in the standard ISO 1996-1, Annex E [10]. The results of this effort indicate that the recommended exposure limit to avoid adverse health effects from aircraft noise should be L_den_ 53 dB.

## Figures and Tables

**Figure 1 ijerph-15-02717-f001:**
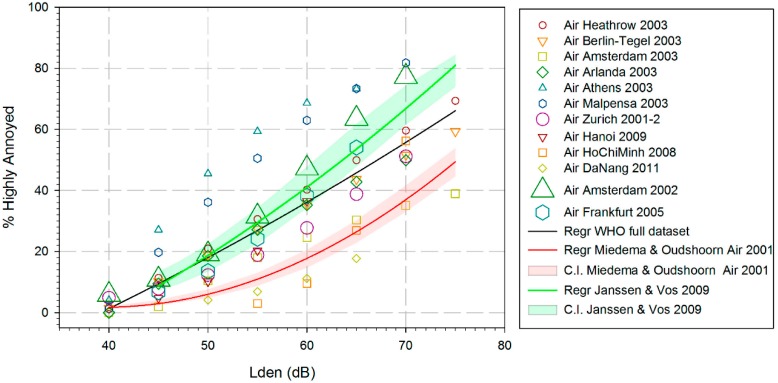
Scatterplot of the response data from the 12 studies included in the WHO full dataset. The size of the markers corresponds to the number of respondents in the respective study.

**Figure 2 ijerph-15-02717-f002:**
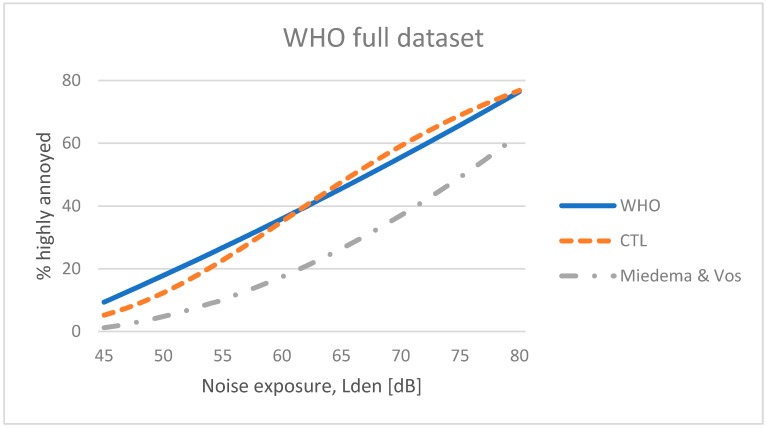
Dose–response curve for the WHO full dataset presented by Guski et al. compared with the corresponding CTL curve and the EU reference curve (Miedema & Vos) for aircraft noise annoyance.

**Figure 3 ijerph-15-02717-f003:**
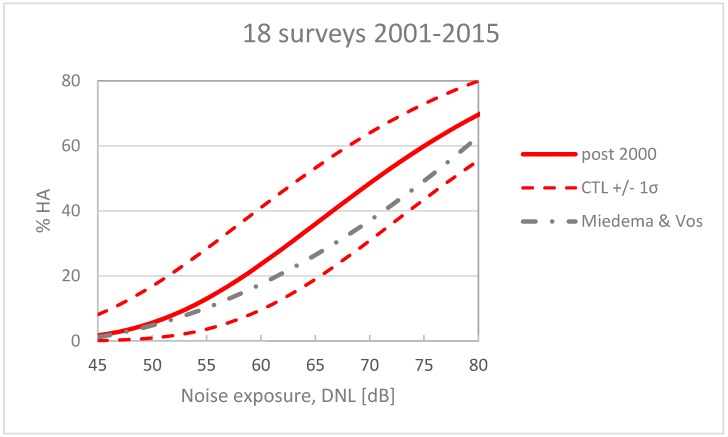
Dose–response curve for 18 post-2000 surveys compared with the EU reference curve (Miedema & Vos) for aircraft noise annoyance.

**Table 1 ijerph-15-02717-t001:** Studies included in the WHO full dataset.

Year	IATA	Airport	Reference	Respondents	CTL	H/L
2003	AMS	Amsterdam	Babisch et al. [4]	898	71.6 dB	H
2003	ATH	Athens	Babisch et al. [4]	635	55.6 dB	H
2003	TXL	Berlin, Tegel	Babisch et al. [4]	972	65.6 dB	L
2003	LHR	Heathrow	Babisch et al. [4]	600	65.0 dB	L
2003	MXP	Milan, Malpensa	Babisch et al. [4]	753	54.6 dB	
2003	ARN	Stockholm	Babisch et al. [4]	1003	67.3 dB	H
2002	AMS	Amsterdam	Breugelmans et al. [5]	5873	63.3 dB	H
2001	ZHR	Zurich	Brink et al. [6]	1816	68.0 dB	
2008	SGN	Ho Chi Minh	Nguyen et al. [7]	880	75.5 dB	L
2009	HAN	Hanoi	Nguyen et al. [7]	824	68.2 dB	L
2011	DAD	Da Nang	Nguyen et al. [8]	528	75.0 dB	L
2005	FRA	Frankfurt	Schreckenberg & Meis [9]	2312	63.3 dB	H

**Table 2 ijerph-15-02717-t002:** Aircraft noise annoyance surveys conducted from 2000 to 2015.

Year	Airport	Reference	Respond	CTL	H/L
2001	ZHR	SWI-525 Brink et al. [6]	1520	68.0	H
2002	AMS	GES-2 Breugelmans et al. [5]	640	63.2	H
2002	MSP	Fidell et al. [21]	495	72.6	L
2003	ZHR	SWI-534 Brink et al. [6]	1444	69.0	H
2003	ANASE	Le Masurier [22]	2132	63.0	L
2005	AMS	GES-3 Breugelmans et al. [5]	478	63.3	H
2005	FRA	Schreckenberg & Meis [9]	2309	63.3	H
2008	SGN	Nguyen et al. [7]	880	75.5	L
2009	HAN	Nguyen et al. [7]	824	68.2	H
2010	CGN	Bartels [15]	1262	67.6	L
2011	DAD	Nguyen et al. [8]	528	75.0	L
2014	BOO	Gjestland et al. [23]	302	81.3	L
2014	TRD	Gjestland et al. [23]	300	82.3	L
2014	HAN	Nguyen et al. [24]	910	65.6	H
2015	OSL	Gjestland et al. [23]	300	68.0	H
2015	SVG	Gjestland et al. [23]	302	80.0	L
2015	TOS	Gjestland et al. [23]	300	83.0	L
2015	HAN	Nguyen et al. [24]	1121	63.0	H

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
