# Peer review of "A Systematic Review of the Basis for WHO’s New Recommendation for Limiting Aircraft Noise Annoyance"

_ijerph, 2018, doi:10.3390/ijerph15122717_

Round 1

Reviewer 1 Report

This is an interesting paper which discusses the limitations of the basis for the new WHO guidelines with respect to aircraft noise annoyance. In particular, this paper indicates that the selection of the aircraft annoyance surveys included in the WHO dataset has been made in violation of some assumed inclusion criteria. It has been also shown that a selection of different surveys based on an identical selection protocol, yields different results.

Generally, it is well elaborated paper and needs only minors updates. It is recommended  to:

-      add a definition of Community Tolerance Level and its abbreviation (see lines 58-62),

-      draw the scale markers on the abscissa in Figs. 2 and 3.

Author Response

I am pleased to see that my paper has been recommended for publication.

The issue about definition of the Community Tolerance Level has been resolved by incerting a foot-note and a reference to the ISO 1996-1

Unfortunately the figures have been generated using a standard template and the introduction of scale markers would mean a major re-editing.

Reviewer 2 Report

General comments:

This paper is interesting showing that the choice of the selected airports has consequences on recommandations of the WHO guide for the threshold noise level associated to health problems.

My only concern is that the style of the text which shows that the author is quite angry. This deserves his statement, because readers could believe that the paper is a settling of scores. But the author's point of view worths to be published. So I propose to improve the text in a more  "neutral" style, avoiding aggressive words.

Abstract:

line 13. Avoid the term "arbitrary" because the selection of the studies are not arbitrary for the WHO gideline authors. I would orientate the conclusion on the fact that the choice of selected airports has a great impact on the threshold value for the recommandations. But for that, this threshold should be calculated by the author of this paper, not only CTL values. This should not be very difficult, because the link between CTL value (for 50% of annoyance) and the level for 10% is quite easy to calculate. If I am not wrong, this value equals to 33.3*log(A) - 12. As the CTL equals to 33.3*log(A) - 5.3, there is just a shift of -17 between CTL and 10% threshold.

Introduction:

line 34. "is based on very imperfect and faulty set of data". I would write: is based on specific set of data whose choice has great impact on the porposed recommandations.

Systematic review:

lines 39-42. Guski is often cited because he is the first author of the previous paper on annoyance. But this is not really clear at a first reading, so I propose to write these sentences in a different way:

The group of researchers who worked on the impact of environmental noise on annoyance published a systematic review of studies conducted from 2000 and onwards [2]. Guski and his co-authors had developed ...

line 50. The reason for excluding studies on the Guski's paper is clearly written on their §3.1.2: the authors need to have a model that they call ERF (exposure response function) and 3 of the 15 studies had only ERR (exposure response relation). This is the reason why they excluded theses 3 airports. So it is not unclear. But the author of this paper could regret it and should propose to keep them or some of them like he does in his Table 2 for Bartels. Why the other two excluded airports are not included in the Table 2?

line 77-78. I would avoid the term "outdated" in order to stay relatively "neutral" towards the concern colleagues.

line 88. I completely agree with the fact that the non acoustic factors are overlooked in the WHO document. I even saw the following sentense which is wrong: p.14 of the WHO guide it is written: "in noise annoyance studies, non acoustic factors may explain up to 33% of the variance"! Is it a Freudian slip?

lines 115-117. The way these sentenses could be interpreted is not clear. I think the interpretation could be developped for each sentense in order to make it clearer that the answers could really differ.

Paragraph 2.4 lines 170-207. The author points out that the High-rate change airports are overrepresented compared to Low-rate airports. But taking into account that majority of airports will have to increase the number of aircraft movements in the future, is it a real drawback? But of course, this should be explained and debated.

Paragraph 2.5:

line 215. What do you mean by "non-standardize" test results? Do it refers to the pargraph 2.2 where the interpretations of the questioins are discussed?

Figure 2:

This Figure presents the calculated CTL for the WHO selection. I doubt that the 50% of the HA persons is at 66. Visually it seams that the red curve has a point of inflexion around 60 and not 66.

In order to give scientific validation of the CTL compared to the WHO model, the explained variance should be calculated. As I think it is not possible to calculate it with CTL, the correlation between mean data and modeled data could be calculated for both curves. Guski said that his model explained 70% of the variance (determination coefficient of 0.7 on mean values). Maybe the correlation coefficients for WHO and CTL could be compared?

Moreover, I think it is important to focus on the 10%HA value, because it is "the" value that is chosen by WHO for the recommandation value. The author should show that changing the way to calculate this 10%HA, using a different approach, has an impact on this value. He/she sould calculate it of course (66.1 - 17.3 = 48.8 around 4dB more than 45dB).

Paragraph 2.6:

Figure 3:

The CTL - sigma could be drawn. That would show that the WHO model is over the mean CTL whereas the Miedema curve is below.

Again, I think the focus sould be on the 10%HA value. Here with this alternative selection of airports, the 10%HA value should be around 70.7 - 17.3 = 53.4 dBLden, which is very close to the other transportation thresholds for the WHO guide (road traffic and railway).

Conclusion:

lines 271-278. This should be rewritten in a more softer style because again, this style does not serve to the paper. The author should not show any emotional reaction against the authors of the guide, but only should show that the selection of the airports, and the chosen method of averaging, have an important impact on the final threshold value. The author of this paper can clearly regret that this fact was not enough clear in the WHO guide.

Author Response

General comments:

This paper is interesting showing that the choice of the selected airports has consequences on recommandations of the WHO guide for the threshold noise level associated to health problems.

My only concern is that the style of the text which shows that the author is quite angry. This deserves his statement, because readers could believe that the paper is a settling of scores. But the author's point of view worths to be published. So I propose to improve the text in a more  "neutral" style, avoiding aggressive words.

I am pleased to see that the reviewer finds my paper worth to be published, and I am exceptionally grateful for the very detailed suggestions for improvement.

I can confirm that the paper was written in a state of desperation when I realized that none of my concerns that had been conveyed to the Guidelines development Group, had been considered in the final version of the Guidelines.

As suggested I have now tried to present the paper in a very neutral form.

Abstract:

line 13. Avoid the term "arbitrary" because the selection of the studies are not arbitrary for the WHO gideline authors. I would orientate the conclusion on the fact that the choice of selected airports has a great impact on the threshold value for the recommandations. But for that, this threshold should be calculated by the author of this paper, not only CTL values. This should not be very difficult, because the link between CTL value (for 50% of annoyance) and the level for 10% is quite easy to calculate. If I am not wrong, this value equals to 33.3*log(A) - 12. As the CTL equals to 33.3*log(A) - 5.3, there is just a shift of -17 between CTL and 10% threshold.

The abstract has been re-written concentrating on the fact that different survey datsets may yield very different results. I have not included any calculations (or numbers) in the abstract, just focusing on the dependency on survey selection and analyzis methods.

(I will also comment that this reviewer is among the few that has fully understood the CTL method and the simplicity of the analysis)

Introduction:

line 34. "is based on very imperfect and faulty set of data". I would write: is based on specific set of data whose choice has great impact on the porposed recommandations.

Noted and re-phrased. "Improper" adjectives removed

Systematic review:

lines 39-42. Guski is often cited because he is the first author of the previous paper on annoyance. But this is not really clear at a first reading, so I propose to write these sentences in a different way:

The group of researchers who worked on the impact of environmental noise on annoyance published a systematic review of studies conducted from 2000 and onwards [2]. Guski and his co-authors had developed ...

Reviewer's suggestion followed

line 50. The reason for excluding studies on the Guski's paper is clearly written on their §3.1.2: the authors need to have a model that they call ERF (exposure response function) and 3 of the 15 studies had only ERR (exposure response relation). This is the reason why they excluded theses 3 airports. So it is not unclear. But the author of this paper could regret it and should propose to keep them or some of them like he does in his Table 2 for Bartels. Why the other two excluded airports are not included in the Table 2?

I have also been notified by Guski regarding the reason for excluding the Trondheim study. In my correspondance  with Guski in connection with his data collection, I tried to point out that the CTL value for this airport  uniquely defines an EER, but Guski does not seem to have understood the CTL method. In order to do a thorough CTL analysis you need the %HA and the number of respondents per exposure bin. This information was not available for the remaining two studies

line 77-78. I would avoid the term "outdated" in order to stay relatively "neutral" towards the concern colleagues.

Noted and re-phrased

line 88. I completely agree with the fact that the non acoustic factors are overlooked in the WHO document. I even saw the following sentense which is wrong: p.14 of the WHO guide it is written: "in noise annoyance studies, non acoustic factors may explain up to 33% of the variance"! Is it a Freudian slip?

Thank you for commenting on this issue. I have included a remark regarding the mistake made by the GDG

lines 115-117. The way these sentenses could be interpreted is not clear. I think the interpretation could be developped for each sentense in order to make it clearer that the answers could really differ.

Issue resolved by introduciing an extra explanation in the following paragraph

Paragraph 2.4 lines 170-207. The author points out that the High-rate change airports are overrepresented compared to Low-rate airports. But taking into account that majority of airports will have to increase the number of aircraft movements in the future, is it a real drawback? But of course, this should be explained and debated.

The HRC/LRC issue can be considered independent of airport expansion. An LRC airport can see an increase in movements or change in aircraft fleet, but the change is gradual over time, and not abrupt as for an HRC airport

Paragraph 2.5:

line 215. What do you mean by "non-standardize" test results? Do it refers to the pargraph 2.2 where the interpretations of the questioins are discussed?

Re-phrased. I meant to say "Test results from a survey not conducted according to standardized recommendations

Figure 2:

This Figure presents the calculated CTL for the WHO selection. I doubt that the 50% of the HA persons is at 66. Visually it seams that the red curve has a point of inflexion around 60 and not 66.

In order to give scientific validation of the CTL compared to the WHO model, the explained variance should be calculated. As I think it is not possible to calculate it with CTL, the correlation between mean data and modeled data could be calculated for both curves. Guski said that his model explained 70% of the variance (determination coefficient of 0.7 on mean values). Maybe the correlation coefficients for WHO and CTL could be compared?

Moreover, I think it is important to focus on the 10%HA value, because it is "the" value that is chosen by WHO for the recommandation value. The author should show that changing the way to calculate this 10%HA, using a different approach, has an impact on this value. He/she sould calculate it of course (66.1 - 17.3 = 48.8 around 4dB more than 45dB).

The data in the figure is correct and has been double-checked and compard with the data in ISO 1996.

Calculating the confidence interval for a set of CTL curves is quite an elaborate task. I am afraid an explanation of how that can be done, will move the focus away from the main message in the text. (I am preparing another paper that is specifically focusing on the use of the CTL method, and an explanation is attached at the end of this rebuttal.

I have also chosen not to do any further calculation of CTL for the dataset provided by Guski et al. A more detailed discussion around theis results would imply a de facto acceptance of their selection methods. In my view the HYENA results for instance should never have been included at all due to the limited age range and use of non-standardized questions.

I have included a new paragraph (line 301) to show the 10% HA level using the new dataset.

Paragraph 2.6:

Figure 3:

The CTL - sigma could be drawn. That would show that the WHO model is over the mean CTL whereas the Miedema curve is below.

Again, I think the focus sould be on the 10%HA value. Here with this alternative selection of airports, the 10%HA value should be around 70.7 - 17.3 = 53.4 dBLden, which is very close to the other transportation thresholds for the WHO guide (road traffic and railway).

The +/- sigma lines have been included

Conclusion:

lines 271-278. This should be rewritten in a more softer style because again, this style does not serve to the paper. The author should not show any emotional reaction against the authors of the guide, but only should show that the selection of the airports, and the chosen method of averaging, have an important impact on the final threshold value. The author of this paper can clearly regret that this fact was not enough clear in the WHO guide.

The conclusion has been completely re-written only focusing on the importance of survey selection and analysis method.

Below is a description of CI calculations for the CTL method

*******************************

CTL maximum likelihood calculation

The maximum likelihood estimator (MLE) is used to determine the lateral position of the CTL curve such that the joint probability of the data set members arising from the CTL function is maximized.  The operative phrase is “joint probability.”  An iterative search procedure is used to determine the CTL value which maximizes this likelihood.

In this particular analysis “likelihood” is synonymous with “probability.”  For each trial value of CTL the probability of each data point’s high annoyance arising from the predictive equation given the data point’s DNL is calculated.  The calculation uses the binomial probability equation:

P(k) =    ( n ) pk qn-k

k

where:

P = probability of k respondents out of n being highly annoyed

k = number of highly annoyed respondents

n = total number of respondents

n – k = number of respondents not highly annoyed

q = 1 – p = probability of not being highly annoyed

The probabilities for each data point are multiplied together to form the joint probability.

Note that the above equation requires the actual numbers of annoyed and total respondents associated with the data point, not the probability that would be calculated as k/n.

CTL confidence interval calculation

Finding the most likely value of CTL (above) involves searching for the peak of a likelihood (probability) versus CTL curve, an empirical probability density function with a bell-shaped appearance.  The cumulative area under this curve is used to estimate confidence intervals.  The probability density curve will not necessarily be perfectly symmetric.  Hence, the confidence interval limits will not necessarily be equidistant from the maximum likelihood value.  The degree of symmetry depends on a number of factors including the number of respondents underlying each data point and the proximity of the percent highly annoyed values to an asymptotic value of zero.

The algorithm employed in the present spreadsheet calculates the probability density function at sufficiently closely spaced intervals of CTL such that this function can be approximated by a series of straight line segments.  Hence, the cumulative area at any point on the curve can be determined by summing the areas of consecutive trapezoids defined by the empirical line segment function (or fractions of a line segment).

Once the TOTAL area under the curve has been established, any percentage of this area can be calculated.  For example, if the 95% confidence interval on the derived CTI value is desired then the CTL values where 2.5% and 97.5% (97.5 – 2.5 = 95.0) of the cumulative area lie are used to establish the confidence interval bounds.  The same method would be used for any desired confidence interval by determining the points at which (100-CI)/2 and 100 – (100-CI)/2 percent of the area lie.

Round 2

Reviewer 2 Report

I thank the author for all the changes he did for improvement.

And thanks also for CI calculations, I will have a look as soon as possible.